# Heart failure treatment patterns: A pharmacoepidemiological descriptive study in Colombia (The HEATCO study)

Manuel E. Machado-Duque[1,2], Andrés Gaviria-Mendoza[1,2],
Luis F. Valladales-Restrepo[1,2], Juan Sebastián Franco[3], María de Rosario Forero[3],
David Vizcaya[4], Jorge E. Machado-Alba[1]*

1 Grupo de Investigación en Farmacoepidemiología y Farmacovigilancia. Universidad Tecnológica de Pereira - Audifarma S.A. Pereira, Colombia, 2 Grupo de investigación Biomedicina. Institución Universitaria Visión de las Américas, Pereira, Colombia, 3 Medical Affairs, Bayer S.A. Bogotá, Colombia, 4 Medical Affairs, Bayer S.A. Barcelona, España

* machado@utp.edu.co

## Abstract

### Introduction

Heart failure is a common condition associated with significant mortality. *Objective*: to determine the prescription patterns of medications for the treatment of heart failure in a cohort of patients from Colombia.

### Methods

This was a retrospective study based on the clinical records of patients diagnosed with heart failure between 2019 and 2020. Sociodemographic, clinical, paraclinical, and pharmacological variables and the specialty of the treating physician were identified. Patients were classified according to functional class, stage, and left ventricular ejection fraction (LVEF).

### Results

A total of 4742 patients were evaluated, with a mean age of 68.2 ± 13.8 years and a male predominance (61.3%). A total of 92.0% were classified as stage C and 54.8% as functional class I, the mean LVEF was 42.9 ± 14.8%, and 32.53% had reduced LVEF. 30.7% did not have LVEF data. The most common causes were ischemic heart disease (44.0%) and arterial hypertension (29.7%). A total of 5.2% had hospitalizations for heart failure in the last year, and 75.6% were attended by a general practitioner. These patients were treated with β-blockers (88.3%), renin-angiotensin-aldosterone system inhibitors (RAASis) (83.1%), loop diuretics (46.8%), and mineralocorticoid receptor antagonists (MRAs) (46.5%). Triple therapy with RAA-Sis + β-blockers+MRAs was received by 56.4% of patients with reduced LVEF, 32.8%

**Data availability statement:** data available at https://www.protocols.io/private/520BB3503DB111F0803E0A58A9FEAC02.

**Funding:** This work was supported by Bayer AG, (Bogotá, Colombia).

**Competing interests:** Manuel Machado-Duque have a contractual relationship with Audifarma SA and Institución Universitaria Visión de las Américas. Andres Gaviria-Mendoza have a contractual relationship with Audifarma SA and Institución Universitaria Visión de las Américas. Luis Valladales-Restrepo have a contractual relationship with Audifarma SA and Institución Universitaria Visión de las Américas. Juan-Sebastian Franco are full-time employee of Bayer Colombia. Maria del Rosario Forero are full-time employee of Bayer Colombia. David Vizcaya are full-time employee of Bayer Hispania (Spain). Jorge E. Machado-Alba have a contractual relationship with Universidad Tecnológica de Pereira and Audifarma SA. This does not alter our adherence to PLOS ONE policies on sharing data and materials.

with mildly reduced LVEF and19.5% with preserved LVEF, while quadruple therapy adding a sodium-glucose cotransporter-2 inhibitor (SGLT2i) was given just to 4.6% with reduced LVEF.

## Conclusion

The treatment that patients with heart failure with preserved LVEF is relatively simpler and is closer to the recommendations, while the proportion of indicated therapies according to guidelines is lower among those with reduced LVEF.

## Introduction

Heart failure (HF) is a common condition that is increasingly prevalent worldwide, with an estimated 64 million people suffering from this disease according to reports in 2023 [1]; HF is currently estimated to affect approximately 2% of the world's population, reaching up to 10% among those over 75 years of age [2]. In Colombia, an overall prevalence of 2.3% was estimated for 2018, which would indicate that more than a million Colombian patients had HF [3].

The classification of HF has been given from different points of view, and each of these can be complementary to the others for diagnostic, therapeutic, and follow-up purposes. HF can be classified based on manifestation into New York Heart Association (NYHA) functional classes I to IV, by disease progression into American College of Cardiology (ACC) and American Heart Association (AHA) stages A to D (ACC/AHA), and according to left ventricular ejection fraction (LVEF) (reduced ≤40% and preserved ≥ 50%) (Yancy et al., 2017). Particularly, these classifications are important to establish the treatment for HF because there are drugs, including renin-angiotensin-aldosterone system inhibitors (RAASis), such as angiotensin II receptor blockers (ARBs) and angiotensin-converting enzyme inhibitors (ACEIs), as well as β-blockers, that are indicated in practically all stages of HF [4,5], unlike drugs such as loop diuretics, which are indicated in stage C patients with congestive symptoms, or mineralocorticoid receptor antagonists (MRAs) in the same patient population with the addition of reduced LVEF (HFrEF); similarly, the use of ARBs plus neprilysin inhibitors (ARNIs) is recommended for patients with persistent symptoms [6], which together with other drugs with neurohormonal effects (ARBs, ACEIs, MRAs) can improve survival, especially in cases of HFrEF [7]. The use of other therapies such as ivabradine, digoxin, and oral vasodilators and the use of devices such as cardiac resynchronization in different particular indications must also be recognized [8]. In general, the treatment of HF should be multidisciplinary to achieve the appropriate and indicated use of drugs and proper treatment compliance, Due to current evidence of new therapies with endocrine, renal and cardiac effects [6]. In the Colombian context, a study by Rojas-Sanchez et al. in 2014 reported in a sample of 161 patients that 38.5% used ACEIs, 49.7% used ARBs, 95.0% used β-blockers, 68.9% used spironolactone, 68.9% used furosemide, and 41.6% used digoxin; a drug therapy noncompliance rate of 16.1% was reported [9].

The Colombian health system offers universal coverage to the entire population through two insurance schemes, one contributory or paid by the employer and the worker, and another subsidized by the state for people without the ability to pay, both of which have a benefit plan that includes most drugs required for the treatment of HF. It is important to know the treatment patterns among patients with HF in a larger sample representing different regions of the country, as well as to be able to evaluate the indications for the use of each drug according to the stage and LVEF and the introduction of new therapies, information that is not available and may be of interest to multiple parties in the health system. Therefore, the objective of this study was to determine the prescription patterns of drugs for the treatment of HF in a group of patients affiliated with one healthcare insurer of Colombia between 2019 and 2020.

## Materials and methods

A descriptive, retrospective study was carried out on the prescription patterns of drugs used in the treatment of HF from a population-based drug dispensing database and clinical records. The prescriptions of adult patients diagnosed with HF from June 01, 2019, to May 31, 2020, were analyzed.

The initial information was obtained from the drug dispensing database of Audifarma S.A., the main pharmaceutical manager in Colombia for the delivery of institutional drugs to outpatients and hospital patients, including patients diagnosed with HF, according to the International Classification of Diseases, Tenth Revision (ICD-10), with the dispensing of some medications during the observation period in outpatient context. Subsequently, individuals who were affiliated with one insurer of the contributory scheme were selected (The insurer provided authorization for access to the clinical record), and their clinical records were reviewed for clinical variables of HF and comorbidities. No exclusion criteria were considered for this study.

From the information on the dispensing of drugs to the included population, the data from the electronic clinical records of the patients were obtained by a group of trained physicians. In addition, two researchers consolidated and validated the information for inconsistent data and errors. The following groups of variables were recorded:

- Sociodemographic: Sex, age, city where health care was sought, and department or region of the country (Bogotá-Cundinamarca, Caribbean region, Central region, Eastern region, Pacific region and Amazon-Orinoquía region)

- Diagnosis and comorbidities: Primary and secondary diagnoses of heart failure according to ICD-10 codes (I50.0; I50.1; I50.9; I11.0; I11.9; I13.0; I13.1; I13.2; I13.9; I42.0; I42.1; I42.2; I51.0; I51.1-I51.9). Each patient was identified according to the classification of a) the NYHA, b) the ACC/AHA, and c) reduced or preserved LVEF. Information on comorbidities was obtained during the entire observation period, including the etiology of heart failure.

- Hospitalizations: The number of hospitalizations and visits to the emergency department during the study period was recorded from the outpatient primary care clinical record.

- Prescribing physician: The specialty of the prescribing physician was identified: general practitioner, internist, cardiologist, pulmonologist, or other.

- Symptoms and signs/paraclinical parameters: a) Symptoms and signs included dyspnea, fatigue, lower limb edema, etc. b) Paraclinical parameters included LVEF (reduced (HFrEF) ≤40%, Mildly reduced (HFmrEF) 41–49% and preserved (HFpEF) ≥50%, no data recorded) – (Left ventricular ejection fraction value was obtained from the report in the medical record of the first echocardiogram recorded during the observation period), creatinine and glomerular filtration rate (GFR), which was calculated by CKD-EPI equation.

- Drugs used for HF: The information of drugs used in the treatment of HF was collected, including active substance, mean dose, dosage form and frequency of use by LVEF classification. The defined daily dose (DDD) was used as the unit of measurement for drug use, according to the recommendations of the World Health Organization (WHO).

a. Comedications: The following were identified: a) antidiabetics, b) antihypertensives and diuretics, c) lipid-lowering agents, d) antiulcer agents, e) antidepressants, f) anxiolytics and hypnotics, g) thyroid hormone, h) antipsychotics, i) antiepileptics, j) analgesics and anti-inflammatories, k) bronchodilators and/or inhaled corticosteroids, l) antiplatelet agents, and m) anticoagulants, among others.

## Statistical analysis

The data were analyzed with the statistical package SPSS Statistics, version 28.0 for Windows (IBM, USA). A descriptive analysis was performed with frequencies and proportions expressing qualitative variables and central trend and dispersion measures for quantitative variables. For quantitative variables, normality was initially tested using the Kolmogorov–Smirnov test; for those variables with a normal distribution, means and standard deviations are reported, and for those with asymmetrical distribution, medians and interquartile ranges are reported. A division of the sociodemographic, clinical, and comorbidity characteristics was made according to LVEF. Regarding the use of medications, the average dose, relationship with the daily dose defined by the WHO, and frequency of use by LVEF classification were identified for those prescribed for HF. UpSet library version 0.6.1 in Python was used to generate the combined drug consumption figure [10].

## Bioethical considerations

The protocol was classified according to Resolution 8430/1993 of the Ministry of Health as research without risk. The ethical principles of justice, nonmaleficence, beneficence, and confidentiality established by the Declaration of Helsinki were respected. The protocol was approved by the Bioethics Committee of the Universidad Tecnológica de Pereira (approval number: 12–150221. Date February 16 of 2021). According to Resolution 8430 of the Colombian Ministry of Health, observational research carried out on clinical records does not require informed consent. Date of access to clinical records: December 9 of 2021 to June 30 of 2022. No personal data of any of the research subjects were included. Patient information was completely anonymized.

## Results

### Sociodemographic and clinical characteristics of patients with HF

A total of 4742 patients diagnosed with HF were evaluated, with a mean age of 68.2 ± 13.8 years and a male predominance (61.3%). The median disease progression time at the start of follow-up was 800 days (interquartile range: 208–1792 days). The main regions represented were the Central, Bogotá-Cundinamarca and Caribbean regions. Most of the patients had a reduced LVEF (n = 1533; 32.3%), followed by those with preserved LVEF (n = 1244; 26.3%), and mildly reduced LVEF (n = 506; 10.7%), missing LVEF record data (n = 1459; 30.7%). Table 1 shows the general characteristics of the included patients.

The vast majority of patients in all categories were classified as stage C (AHA/ACC), with a higher percentage of stage B among patients with preserved LVEF than among those with reduced LVEF (8.5% vs. 1.4%). On average, for the entire group analyzed, the LVEF was above 40%. The main etiologies of HF identified were coronary artery disease (44.9%) and arterial hypertension (29.7%). The most frequently identified comorbidity in all subgroups was arterial hypertension, followed by coronary artery disease, obesity, diabetes mellitus, and atrial fibrillation, among others. In addition, approximately one-third of the patients had a GFR of less than 60 mL/min, placing them in groups of stage 3A or more severe chronic kidney disease (see Table 1).

Some symptoms were identified in patients at the start of follow-up, with reports of dyspnea upon slight exertion or worse (n = 1150; 24.3%), angina (n = 682; 14.4%), lower limb edema (n = 583; 12.3%), fatigue (n = 189; 4.0%), and paroxysmal nocturnal dyspnea (n = 103; 2.2%) among the most frequent, but clinical signs such as jugular venous distention (n = 9; 0.2%) and hepatomegaly (n = 7; 0.1%) were also reported.

**Table 1. Sociodemographic, clinical, pharmacological and health care characteristics of 4742 patients diagnosed with HF, from Colombia.**

| Characteristic | Total (n = 4742) | | Preserved [a]LVEF (HFpEF) (n = 1244; 26,3%) | | Mildly reduced LVEF (HFmrEF) (n = 506; 10,7) | | Reduced LVEF (HFrEF) (n = 1533; 32,3%) | | LVEF No data (n = 1459; 30,7%) | |
|---|---|---|---|---|---|---|---|---|---|---|
| | n | % | n | % | n | % | n | % | n | % |
| Age | | | | | | | | | | |
| Mean (SD)[b] | 68.2 (13.8) | | 69.6 (13.7) | | 68.6 (13.3) | | 66.6 (12.9) | | 68.7 (14.7) | |
| Median (IQR)[c] | 69.0 (59.0 - 79.0) | | 71.0 (60.0 - 80.0) | | 69.0 (59.0 - 79.0) | | 67.0 (58.0 - 76.0) | | 69.0 (59.0 - 80.0) | |
| Gender | | | | | | | | | | |
| Male | 2907 | 61.3 | 718 | 57.7 | 315 | 62.3 | 999 | 65.2 | 875 | 60.0 |
| Female | 1835 | 38.7 | 526 | 42.3 | 191 | 37.7 | 534 | 34.8 | 584 | 40.0 |
| Geographic regions | | | | | | | | | | |
| Central | 1438 | 30.3 | 357 | 28.7 | 154 | 30.4 | 473 | 30.9 | 454 | 31.1 |
| Bogota-Cundinamarca | 1271 | 26.8 | 350 | 28.1 | 137 | 27.1 | 444 | 29.0 | 340 | 23.3 |
| Caribbean | 1197 | 25.2 | 322 | 25.9 | 114 | 22.5 | 312 | 20.4 | 449 | 30.8 |
| Oriental | 579 | 12.2 | 152 | 12.2 | 71 | 14.0 | 200 | 13.0 | 156 | 10.7 |
| Pacific | 257 | 5.4 | 63 | 5.1 | 30 | 5.9 | 104 | 6.8 | 60 | 4.1 |
| Body-mass index | | | | | | | | | | |
| Mean (SD) | 27.2 (5.2) | | 27.7 (5.3) | | 27.0 (5.0) | | 26.4 (4.8) | | 27.7 (5.4) | |
| Median (IQR) | 26.7 (23.7 - 30.0) | | 27.3 (24.2 - 30.4) | | 26.4 (23.7 - 29.7) | | 26.0 (23.3 - 29.1) | | 27.0 (24.0 - 30.8) | |
| AHA classification[d] (A.B.C.D) | | | | | | | | | | |
| A | 0 | 0 | 0 | 0.0 | 0 | 0.0 | 0 | 0.0 | 0 | 0.0 |
| B | 318 | 6.7 | 106 | 8.5 | 41 | 8.1 | 21 | 1.4 | 150 | 10.3 |
| C | 4361 | 92.0 | 1133 | 91.1 | 460 | 90.9 | 1470 | 95.9 | 1298 | 89.0 |
| D | 63 | 1.3 | 5 | 0.4 | 5 | 1.0 | 42 | 2.7 | 11 | 0.8 |
| Functional class NYHA[e] | | | | | | | | | | |
| I | 2598 | 54.8 | 691 | 55.5 | 290 | 57.3 | 782 | 51.0 | 835 | 57.2 |
| II | 1376 | 29 | 391 | 31.4 | 149 | 29.4 | 422 | 27.5 | 414 | 28.4 |
| III | 680 | 14.3 | 144 | 11.6 | 63 | 12.5 | 295 | 19.2 | 178 | 12.2 |
| IV | 88 | 1.9 | 18 | 1.4 | 4 | 0.8 | 34 | 2.2 | 32 | 2.2 |
| First LVEF value % | | | | | | | | | | |
| Mean (SD) | 42.9 (14.8) | | 58.6 (5.6) | | 45.2 (2.3) | | 29.4 (7.7) | | NA | |
| Median (IQR) | 43.0 (30.0 - 55.0) | | 59.0 (55.0 - 61.0) | | 45.0 (43.0 - 47.0) | | 30.0 (25.0 - 35.0) | | NA | |
| Comorbidities | | | | | | | | | | |
| Arterial hypertension | 4200 | 88.6 | 1121 | 90.1 | 458 | 90.5 | 1305 | 85.1 | 1316 | 90.2 |
| Coronary artery disease | 2260 | 47.7 | 505 | 40.6 | 297 | 58.7 | 865 | 56.4 | 593 | 40.6 |
| Obesity | 1272 | 26.8 | 365 | 29.3 | 122 | 24.1 | 328 | 21.4 | 457 | 31.3 |
| Diabetes mellitus | 1200 | 25.3 | 297 | 23.9 | 108 | 21.3 | 416 | 27.1 | 379 | 26 |
| Atrial fibrillation | 998 | 21.0 | 276 | 22.2 | 105 | 20.8 | 347 | 22.6 | 270 | 18.5 |
| Chronic Obstructive Pulmonary Disease | 876 | 18.5 | 280 | 22.5 | 76 | 15.0 | 244 | 15.9 | 276 | 18.9 |
| Sleep apnea | 238 | 5.0 | 83 | 6.7 | 29 | 5.7 | 62 | 4.0 | 64 | 4.4 |
| Kidney failure GFR<90 (n=4080)[f] | 3216 | 78.8 | 868 | 77.7 | 347 | 77.1 | 1053 | 80.3 | 948 | 78.9 |
| GFR – mean (SD) | 70.1 (22.7) | | 71.2 (22.4) | | 70.5 (22.4) | | 68.9 (22.7) | | 70.2 (23.2) | |
| Stage 1 (normal) | 864 | 21.2 | 249 | 22.3 | 103 | 22.9 | 259 | 19.7 | 253 | 21.1 |
| Stage 2 | 1924 | 47.2 | 543 | 48.6 | 209 | 46.4 | 604 | 46.0 | 568 | 47.3 |
| Stage 3a | 701 | 17.2 | 164 | 14.7 | 78 | 17.3 | 251 | 19.1 | 208 | 17.3 |
| Stage 3b | 392 | 9.6 | 116 | 10.4 | 37 | 8.2 | 129 | 9.8 | 110 | 9.2 |

*(Continued)*

| Characteristic | Total (n = 4742) | | Preserved ªLVEF | | Mildly reduced LVEF | | Reduced LVEF | | LVEF | |
|---|---|---|---|---|---|---|---|---|---|---|
| | | | (HFpEF) (n = 1244; 26,3%) | | (HFmrEF) (n = 506; 10,7) | | (HFrEF) (n = 1533; 32,3%) | | No data (n = 1459; 30,7%) | |
| | n | % | n | % | n | % | n | % | n | % |
| Stage 4 | 145 | 3.6 | 34 | 3.0 | 18 | 4.0 | 48 | 3.7 | 45 | 3.7 |
| Stage 5 | 54 | 1.3 | 11 | 1.0 | 5 | 1.1 | 21 | 1.6 | 17 | 1.4 |
| No data | 662 | NA | 127 | NA | 56 | NA | 221 | NA | 258 | NA |
| Possible causes/ mechanisms heart failure | | | | | | | | | | |
| Coronary artery disease | 2130 | 44.9 | 470 | 37.8 | 279 | 55.1 | 814 | 53.1 | 567 | 38.9 |
| Arterial hypertension | 1410 | 29.7 | 373 | 30.0 | 121 | 23.9 | 390 | 25.4 | 526 | 36.1 |
| Cardiomyopathies | 766 | 16.2 | 172 | 13.8 | 86 | 17.0 | 285 | 18.6 | 223 | 15.3 |
| Arrythmias | 541 | 11.4 | 157 | 12.6 | 53 | 10.5 | 163 | 10.6 | 168 | 11.5 |
| Valvulopathies | 547 | 11.5 | 199 | 16.0 | 58 | 11.5 | 138 | 9.0 | 152 | 10.4 |
| Infections | 171 | 3.6 | 34 | 2.7 | 16 | 3.2 | 87 | 5.7 | 34 | 2.3 |
| Congenital | 53 | 1.1 | 13 | 1.0 | 3 | 0.6 | 11 | 0.7 | 26 | 1.8 |
| Metabolic and autoimmune | 23 | 0.5 | 8 | 0.6 | 2 | 0.4 | 8 | 0.5 | 5 | 0.3 |
| Adverse drug reaction | 22 | 0.5 | 6 | 0.5 | 0 | 0.0 | 12 | 0.8 | 4 | 0.3 |
| Neuromuscular | 5 | 0.1 | 0 | 0.0 | 0 | 0.0 | 1 | 0.1 | 4 | 0.3 |
| Pericardial disease | 3 | 0.1 | 1 | 0.1 | 0 | 0.0 | 2 | 0.1 | 0 | 0.0 |
| Infiltrative | 1 | 0.0 | 0 | 0.0 | 0 | 0.0 | 1 | 0.1 | 0 | 0.0 |
| Endomyocardial | 1 | 0.0 | 0 | 0.0 | 0 | 0.0 | 1 | 0.1 | 0 | 0.0 |
| Other | 2 | 0.0 | 0 | 0.0 | 0 | 0.0 | 0 | 0.0 | 2 | 0.1 |
| Hospital care | | | | | | | | | | |
| Hospitalizations for any cause | | | | | | | | | | |
| 0 | 4332 | 91.4 | 1155 | 92.8 | 468 | 92.5 | 1386 | 90.4 | 1323 | 90.7 |
| 1 | 345 | 7.3 | 75 | 6.0 | 34 | 6.7 | 129 | 8.4 | 107 | 7.3 |
| 2 or more | 65 | 1.4 | 14 | 1.1 | 4 | 0.8 | 18 | 1.2 | 29 | 2.0 |
| Average hospitalizations (at least one) (SD) | 1.2 (0.6) | | 1.2 (0.4) | | 1.2 (0.6) | | 1.1 (0.4) | | 1.4 (0.9) | |
| Heart failure hospitalizations | | | | | | | | | | |
| 0 | 4467 | 94.2 | 1183 | 95.1 | 491 | 97.0 | 1424 | 92.9 | 1369 | 93.8 |
| 1 | 247 | 5.2 | 51 | 4.1 | 13 | 2.6 | 99 | 6.5 | 84 | 5.8 |
| 2 or more | 28 | 0.6 | 10 | 0.8 | 2 | 0.4 | 10 | 0.7 | 6 | 0.4 |
| Average hospitalizations (at least one) (SD) | 1.1 (0.4) | | 1.2 (0.4) | | 1.2 (0.6) | | 1.1 (0.4) | | 1.1 (0.5) | |
| Emergency consultation for any reason | | | | | | | | | | |
| 0 | 4113 | 86.7 | 1083 | 87.1 | 448 | 88.5 | 1319 | 86.0 | 1263 | 86.6 |
| 1 | 399 | 8.4 | 96 | 7.7 | 41 | 8.1 | 134 | 8.7 | 128 | 8.8 |
| 2 or more | 230 | 4.9 | 65 | 5.2 | 17 | 3.4 | 80 | 5.2 | 68 | 4.7 |
| Average emergencies (at least one) (SD) | 1.7 (1.5) | | 1.8 (1.5) | | 1.6 (1.0) | | 1.8 (1.7) | | 1.7 (1.4) | |
| Emergency visit for heart failure | | | | | | | | | | |
| 0 | 4548 | 95.9 | 1208 | 97.1 | 491 | 97.0 | 1454 | 94.8 | 1395 | 95.6 |
| 1 | 168 | 3.5 | 28 | 2.3 | 13 | 2.6 | 71 | 4.6 | 56 | 3.8 |
| 2 or more | 26 | 0.5 | 8 | 0.6 | 2 | 0.4 | 8 | 0.5 | 8 | 0.5 |
| Average emergencies (at least one) (SD) | 1.2 (0.7) | | 1.3 (0.6) | | 1.1 (0.4) | | 1.2 (1.0) | | 1.2 (0.5) | |
| Medical specialty | | | | | | | | | | |
| General medicine | 3585 | 75.6 | 929 | 74.7 | 356 | 70.4 | 1081 | 70.5 | 1219 | 83.6 |

*(Continued)*

Table 1. (Continued)

| Characteristic | Total (n = 4742) | | Preserved [a]LVEF | | Mildly reduced LVEF | | Reduced LVEF | | LVEF | |
| --- | --- | --- | --- | --- | --- | --- | --- | --- | --- | --- |
| | | | (HFpEF) (n = 1244; 26,3%) | | (HFmrEF) (n = 506; 10,7) | | (HFrEF) (n = 1533; 32,3%) | | No data (n = 1459; 30,7%) | |
| | n | % | n | % | n | % | n | % | n | % |
| Internal medicine | 727 | 15.3 | 217 | 17.4 | 88 | 17.4 | 260 | 17.0 | 162 | 11.1 |
| Cardiology | 304 | 6.4 | 69 | 5.5 | 44 | 8.7 | 152 | 9.9 | 39 | 2.7 |
| Other | 126 | 2.7 | 29 | 2.3 | 18 | 3.6 | 40 | 2.6 | 39 | 2.7 |
| | | | | | | | | | | |
| **Medications** | | | | | | | | | | |
| Number of medications – mean (SD) | 2.9 (1.1) | | 2.6 (1.0) | | 2.9 (1.0) | | 3.4 (1.1) | | 2.8 (1.1) | |
| RAASi [g] | 3940 | 83.1 | 1030 | 82.8 | 437 | 86.4 | 1312 | 85.6 | 1161 | 79.6 |
| ARB[h] | 2295 | 48.4 | 724 | 58.2 | 268 | 53.0 | 532 | 34.7 | 771 | 52.8 |
| ACEi[i] | 1148 | 24.2 | 274 | 22.0 | 121 | 23.9 | 450 | 29.4 | 303 | 20.8 |
| ARNI[j] | 612 | 12.9 | 45 | 3.6 | 61 | 12.1 | 405 | 26.4 | 101 | 6.9 |
| β-blocker | 4187 | 88.3 | 1064 | 85.5 | 468 | 92.5 | 1443 | 94.1 | 1212 | 83.1 |
| Loop diuretic | 2218 | 46.8 | 488 | 39.2 | 202 | 39.9 | 827 | 53.9 | 701 | 48.0 |
| Thiazide diuretic | 541 | 11.4 | 212 | 17.0 | 62 | 12.3 | 79 | 5.2 | 188 | 12.9 |
| Mineralocorticoid receptor antagonist (MRA) | 2205 | 46.5 | 341 | 27.4 | 204 | 40.3 | 1097 | 71.6 | 563 | 38.6 |
| Ivabradine | 124 | 2.6 | 16 | 1.3 | 13 | 2.6 | 77 | 5.0 | 18 | 1.2 |
| Digoxin | 331 | 7.0 | 39 | 3.1 | 22 | 4.3 | 156 | 10.2 | 114 | 7.8 |
| SGLT2i (Flozins) [k] | 256 | 5.4 | 53 | 4.3 | 22 | 4.3 | 108 | 7.0 | 73 | 5.0 |
| Uses RAASi + β-bloquer+MRA | 1654 | 34.9 | 242 | 19.5 | 166 | 32.8 | 865 | 56.4 | 381 | 26.1 |
| Uses RAASi + β-bloquer+MRA+ SGLT2i | 117 | 2.5 | 15 | 1.2 | 8 | 1.6 | 70 | 4.6 | 24 | 1.6 |

[a]LVEF: Left ventricular ejection fraction.

[b]SD: standard deviation.

[c]IQR: interquartile range.

[d]AHA: American Heart Association.

[e]NYHA: New York Heart Association.

[f]% of patients with glomerular filtration rate-GFR data n = 4080 and according to LVEF categories (preserved n = 1727. reduced n = 1152 and without data n = 1201).

[g]RAASi: Renin angiotensin aldosterone system inhibitors.

[h]ARB: Angiotensin II receptor blocker.

[i]ACEi: Angiotensin converting enzyme inhibitor.

[j]ARNI: Angiotensin Receptor-Neprilysin Inhibitor.

[k]SGLT2i: Sodium Glucose Cotransporter 2 inhibitor.

Regarding hospitalizations and emergency department visits due to HF during the observation period, there were similar results in the preserved mildly reduced and reduced LVEF groups (Table 1). However, proportionally, 5.2% (n = 51) of those with preserved LVEF had one or more hospitalizations for HF, compared to 6.5% (n = 99) of people with reduced LVEF. Similarly, 2.3% (n = 28), 2.6% (n = 13) and 5.2% (n = 71) patients visited the emergency department for HF in each group, respectively. Finally, most of the patients were managed by general medicine practitioners, followed by internists and cardiologists.

## Pharmacological management of HF

Regarding pharmacological management, most of the patients were on β-blockers (88.3%), particularly carvedilol (61.4%), and 83.1% were on RAASis. In this last group, the most commonly used were ARBs (48.4%), with a lower proportion of ACEIs and ARNIs (sacubitril-valsartan) used. Triple therapy with β-blockers, RAASis, and MRAs was observed in 34.9%, whereas quadruple therapy adding a sodium-glucose cotransporter-2 inhibitor (SGLT2i) was identified in only 2.5% of patients. Table 2 shows the drugs used for HF by group, mean dose, range, and category of recorded LVEF. Homogeneity was found in the selection of specific drugs within each class, with the predominant use of enalapril among ACEIs, losartan among ARBs, spironolactone among MRAs, etc. In addition, Fig 1 shows the main drug combinations according to frequency of use, the most common being the combination of β-blockers + ARBs + loop diuretics. A total of 24 patients (0.5%) who did not receive any specific therapy for the management of their cardiac condition were identified.

When reviewing pharmacological management according to LVEF, in patients with reduced LVEF, a high use of β-blockers (94.1%) and RAASis (85.6%) was maintained, with greater use of ARNIs (26.4%). Similarly, in this group of patients, a greater use of MRA was identified (71.6%), particularly spironolactone, as well as loop diuretics (53.9%). Triple and quadruple therapy was also used more frequently in this group, at proportions of 56.4% and 4.6%, respectively. In patients with preserved LVEF, the use of β-blockers (88.3%) and RAASis (83.1%) was maintained, with a higher proportion of thiazide diuretics (17.0%) and less use of MRAs (27.4%) compared with patients with reduced LVEF. And in patients with mildly reduced LVEF a higher proportion of use of RAASi (86.4%) than in the other groups, and more frequent use of β-blockers (92.5%) compared to those with preserved LVEF and less frequent than those with reduced LVEF. In general, SGLT2is were used infrequently, with empagliflozin being the most common. The most frequently used concomitant medication was part of the statin group, followed by acetylsalicylic acid, proton pump inhibitors, anticoagulants, and vasoselective calcium channel blockers, but there was a great diversity of drugs that accompanied the management of this group of patients and their comorbidities (see Table 3).

## Discussion

Patients with heart failure with preserved ejection fraction (HFpEF) and mildly reduced ejection fraction (HFmrEF), but mainly those with reduced ejection fraction (HFrEF) have high mortality, suffer frequent hospitalizations and, in addition, their quality of life is seriously affected; therefore, the use of drugs with a positive impact on these outcomes constitutes a critical point in long-term therapy [7,11]. Thus, the identification and clinical classification of this group of patients with HF in Colombia, which established treatment patterns categorized by different variables such as clinical and functional classification, mean LVEF value, comorbidities, and even region of the country, provides relevant information for the identification of patients who are receiving management that is not optimal, insufficient or inappropriate and creates the opportunity to improve the prescription of medications.

The study published in 2021 by the RECOLFACA group in Colombia evaluated 2528 patients with HF and found that there was a male predominance and a median age similar to that of this analysis [12]; however, unlike this previous study, it was possible to identify the use of new drugs such as ARNIs and SGLT2is that have recently been incorporated into clinical practice guidelines [13–15]. The results related to age and sex were also consistent with the findings of the study by Störk et al. in Germany, who reported a male predominance (65.9%) and a mean age of 68 years from a database with a record of more than 120,000 patients with HF [16], And this is consistent with the report from the INTER-CHF study with 5,823 patients worldwide, of whom 9% were from Latin America, with an average age of 67%, but different from Africa with 53%, and maintaining a male predominance [17].In Italy, Rea et al. found a median age of 76 years, with those with HFpEF [18] being approximately three years older, which is higher than that identified in the Colombian patient groups, which can be explained by differences in population, health systems, access to more timely treatment for coronary syndromes and interventions for different risk factors; this study reported that the Hispanic population has a higher frequency

**Table 2. Pharmacological treatment received by 4742 patients in ambulatory setting diagnosed with HF. From Colombia.**

| Characteristic | Total (n=4742) | | Average dose (SD)[a] – mg/day | Average daily dose ratio/ DDD[b] | Min | Max | Median | Preserved LVEF[c] (HFpEF) (n=1244) | | | Mildly reduced LVEF (HFmrEF) (n=506) | | | Reduced LVEF (HFrEF) (n=1533) | | | No data LVEF (n=1459) | | |
|---|---|---|---|---|---|---|---|---|---|---|---|---|---|---|---|---|---|---|---|
| | n | % | | | | | | n | % | Average dose – mg/day | n | % | Average dose – mg/day | n | % | Average dose – mg/day | n | % | Average dose – mg/day |
| **Medications** | | | | | | | | | | | | | | | | | | | |
| ACEi[d] | 1148 | 24.2 | | | | | | 274 | 22.0 | | 121 | 23.9 | | 450 | 29.4 | | 303 | 20.8 | |
| Enalapril | 1125 | 23.7 | 17.1 (13.9) | 1.7 | 2.5 | 40 | 10 | 264 | 21.2 | 18.9 | 120 | 23.7 | 16.9 | 445 | 29.0 | 15.6 | 296 | 20.3 | 17.9 |
| Captopril | 10 | 0.2 | 82.5 (44.2) | 1.7 | 25 | 150 | 75 | 2 | 0.2 | 100.0 | 1 | 0.2 | 50.0 | 3 | 0.2 | 116.7 | 4 | 0.3 | 56.2 |
| Perindopril | 10 | 0.2 | 6.8 (2.8) | 1.7 | 4 | 10 | 5 | 7 | 0.6 | 7.7 | 0 | 0.0 | 0.0 | 2 | 0.1 | 4.5 | 1 | 0.1 | 5.0 |
| Ramipril | 2 | 0.0 | 7.5 (3.5) | 3.0 | 5 | 10 | 7.5 | 0 | 0.0 | 0.0 | 0 | 0.0 | 0.0 | 0 | 0.0 | 0.0 | 2 | 0.1 | 7.5 |
| Lisinopril | 1 | 0.0 | 5.0 | 0.5 | | | | 1 | 0.1 | 5.0 | 0 | 0.0 | 0.0 | 0 | 0.0 | 0.0 | 0 | 0 | 0.0 |
| ARB[e] | 2295 | 48.4 | | | | | | 724 | 58.2 | | 268 | 53.0 | | 532 | 34.7 | | 771 | 52.8 | |
| Losartan | 1978 | 41.7 | 90.4 (31.2) | 1.8 | 25 | 200 | 100 | 628 | 50.5 | 94.0 | 221 | 43.7 | 93.9 | 460 | 30.0 | 83.9 | 669 | 45.9 | 90.5 |
| Valsartan | 128 | 2.7 | 211.9 (93.7) | 2.6 | 40 | 320 | 160 | 38 | 3.1 | 240.0 | 16 | 3.2 | 210.0 | 30 | 2.0 | 173.3 | 44 | 3 | 214.5 |
| Candesartan | 78 | 1.6 | 16.6 (10.8) | 2.1 | 4 | 64 | 16 | 13 | 1.0 | 19.7 | 20 | 4.0 | 15.4 | 27 | 1.8 | 15.0 | 18 | 1.2 | 18.0 |
| Irbesartan | 65 | 1.4 | 251.5 (70.7) | 1.7 | 150 | 0 | 0 | 30 | 2.4 | 265.0 | 3 | 0.6 | 250.0 | 9 | 0.6 | 233.3 | 23 | 1.6 | 241.3 |
| Telmisartan | 33 | 0.7 | 77.6 (26.3) | 1.9 | 40 | 160 | 80 | 12 | 1.0 | 90.0 | 7 | 1.4 | 68.6 | 5 | 0.3 | 72.0 | 9 | 0.6 | 71.1 |
| Olmesartan | 12 | 0.3 | 38.3 (15.9) | 1.9 | 20 | 80 | 40 | 3 | 0.2 | 26.7 | 1 | 0.2 | 40.0 | 0 | 0.0 | 0.0 | 8 | 0.5 | 42.5 |
| Eprosartan | 1 | 0.0 | 600.0 | 1.0 | | | | 0 | 0.0 | 0.0 | 0 | 0.0 | 0.0 | 1 | 0.1 | 600.0 | 0 | 0 | 0.0 |
| ARNI (sacubitril-valsartan)[f] | 612 | 12.9 | 144.2 (94.4) | 0.4 | 50 | 400 | 100 | 45 | 3.6 | 151.1 | 61 | 12.1 | 142.5 | 405 | 26.4 | 141.3 | 101 | 6.9 | 153.5 |
| β-blocker | 4187 | 88.3 | | | | | | 1064 | 85.5 | | 468 | 92.5 | | 1443 | 94.1 | | 1212 | 83.1 | |
| Carvedilol | 2913 | 61.4 | 22.2 (15.1) | 0.6 | 3.1 | 75 | 12.5 | 639 | 51.4 | 21.8 | 322 | 63.6 | 23.5 | 1107 | 72.2 | 23.4 | 845 | 57.9 | 20.2 |
| Metoprolol | 991 | 20.9 | 92.4 (46.9) | 0.6 | 25 | 300 | 100 | 330 | 26.5 | 88.4 | 101 | 20.0 | 103.5 | 249 | 16.2 | 94.4 | 311 | 21.3 | 91.5 |
| Bisoprolol | 229 | 4.8 | 6.1 (3.1) | 0.6 | 1.3 | 10 | 5 | 77 | 6.2 | 6.3 | 34 | 6.7 | 6.4 | 75 | 4.9 | 5.9 | 43 | 2.9 | 6.0 |
| Nebivolol | 48 | 1.0 | 6.3 (3.8) | 1.3 | 2.5 | 25 | 5 | 13 | 1.0 | 6.0 | 11 | 2.2 | 6.4 | 12 | 0.8 | 4.6 | 12 | 0.8 | 8.5 |
| Propranolol | 6 | 0.1 | 63.3 (36.7) | 0.4 | 20 | 120 | 60 | 5 | 0.4 | 60.0 | 0 | 0.0 | 0.0 | 0 | 0.0 | 0.0 | 1 | 0.1 | 80.0 |
| Loop diuretic | 2218 | 46.8 | 41.7 (11.2) | 1.0 | 10 | 200 | 40 | 488 | 39.2 | 40.8 | 202 | 39.9 | 41.1 | 827 | 53.9 | 42.5 | 701 | 48 | 41.7 |
| Thiazide diuretic | 541 | 11.4 | | | | | | 212 | 17.0 | | 62 | 12.3 | | 79 | 5.2 | | 188 | 12.9 | |

*(Continued)*

Table 2. (Continued)

| Characteristic | Total (n=4742) | | Average dose (SD)[a] – mg/day | Average daily dose ratio/ DDD[b] | Min | Max | Median | Preserved LVEF[c] (HFpEF) (n=1244) | | | Mildly reduced LVEF (HFmrEF) (n=506) | | | Reduced LVEF (HFrEF) (n=1533) | | | No data LVEF (n=1459) | | |
|---|---|---|---|---|---|---|---|---|---|---|---|---|---|---|---|---|---|---|---|
| | n | % | | | | | n | % | Average dose – mg/day | n | % | Average dose – mg/day | n | % | Average dose – mg/day | n | % | Average dose – mg/day | |
| Hydrochlorthiazide | 505 | 10.6 | 23.4 (4.2) | 0.9 | 12.5 | 25 | 25 | 198 | 15.9 | 23.8 | 55 | 10.9 | 22.7 | 73 | 4.8 | 22.9 | 179 | 12.3 | 23.2 |
| Indapamide | 33 | 0.7 | 1.97 (0.7) | 0.8 | 1.5 | 5 | 1.5 | 14 | 1.1 | 2.1 | 4 | 0.8 | 2.6 | 6 | 0.4 | 1.5 | 9 | 0.6 | 1.8 |
| Chlorthalidone | 3 | 0.1 | 16.7 (7.2) | 0.7 | 12.5 | 25 | 12.5 | 0 | 0.0 | 0.0 | 3 | 0.6 | 16.7 | 0 | 0.0 | 0.0 | 0 | 0 | |
| Mineralocorticoid Receptor Antagonist (MRA) | 2205 | 46.5 | | | | | | 341 | 27.4 | | 204 | 40.3 | | 1097 | 71.6 | | 563 | 38.6 | |
| Spironolactone | 2090 | 44.1 | 26.0 (9.2) | 0.3 | 12.5 | 150 | 25 | 328 | 26.4 | 25.6 | 191 | 37.7 | 25.4 | 1032 | 67.3 | 25.8 | 539 | 36.9 | 26.6 |
| Eplerenone | 115 | 2.4 | 26.8 (7.6) | 0.5 | 12.5 | 50 | 25 | 13 | 1.0 | 29.8 | 13 | 2.6 | 25.0 | 65 | 4.2 | 26.8 | 24 | 1.6 | 26.0 |
| Ivabradine | 124 | 2.6 | 9.9 (3.4) | 1.0 | 2.5 | 17.5 | 10 | 16 | 1.3 | 10.0 | 13 | 2.6 | 10.6 | 77 | 5.0 | 9.6 | 18 | 1.2 | 10.9 |
| Digoxin | 331 | 7.0 | 0.23 (0.22) | 0.9 | 0.1 | 1 | 0.1 | 39 | 3.1 | 0.2 | 22 | 4.3 | 0.2 | 156 | 10.2 | 0.3 | 114 | 7.8 | 0.2 |
| SGLT2i (Flozins)[g] | 256 | 5.4 | | | | | | 53 | 4.3 | | 22 | 4.3 | | 108 | 7.0 | | 73 | 5 | |
| Empagliflozin | 213 | 4.5 | 19.8 (7.4) | 1.1 | 10 | 50 | 25 | 45 | 3.6 | 20.0 | 17 | 3.4 | 24.7 | 88 | 5.7 | 19.3 | 63 | 4.3 | 19.0 |
| Dapagliflozin | 42 | 0.9 | 9.8 (1.1) | 1.0 | 5 | 10 | 10 | 8 | 0.6 | 10.0 | 5 | 1.0 | 10.0 | 19 | 1.2 | 10.0 | 10 | 0.7 | 9.0 |
| Canagliflozin | 1 | 0.0 | 100.0 | 0.5 | | | | 0 | 0.0 | 0.0 | 0 | 0.0 | 0.0 | 1 | 0.1 | 100.0 | 0 | 0 | 0.0 |

[a]SD: standard deviation.

[b]DDD: defined daily dose.

[c]LVEF: left ventricular ejection fraction.

[d]ACEI: angiotensin converting enzyme inhibitors.

[e]ARB: angiotensin II receptor antagonists.

[f]ARNI: Angiotensin Receptor-Neprilysin Inhibitor.

[g]SGLT2i: Sodium Glucose Cotransporter 2 inhibitor. Note: Of the SGLT2i. 28 were used specifically for heart failure. the rest for diabetes.

of obesity and adverse cardiovascular outcomes, representing a greater risk of developing HF earlier [19,20], which can be added to the recently published analysis based on information from 40 countries in which the differences in the causes of HF were identified between high-income and lower-income regions, showing a higher proportion of coronary disease in high-income and medium-to-high-income countries. Likewise, a better opportunity for care and lower mortality due to complications of HF in the first 30 days after hospitalization were identified, highlighting the importance of timely management and access to recommended therapies [21]. An additional point to keep in mind is that these older adult patients, with multiple comorbidities, are at high risk of presenting frailty from a multidimensional perspective, as they are affected by reduced physical activity, asthenia, decreased gait and speed, and muscle weakness, all directly or indirectly associated

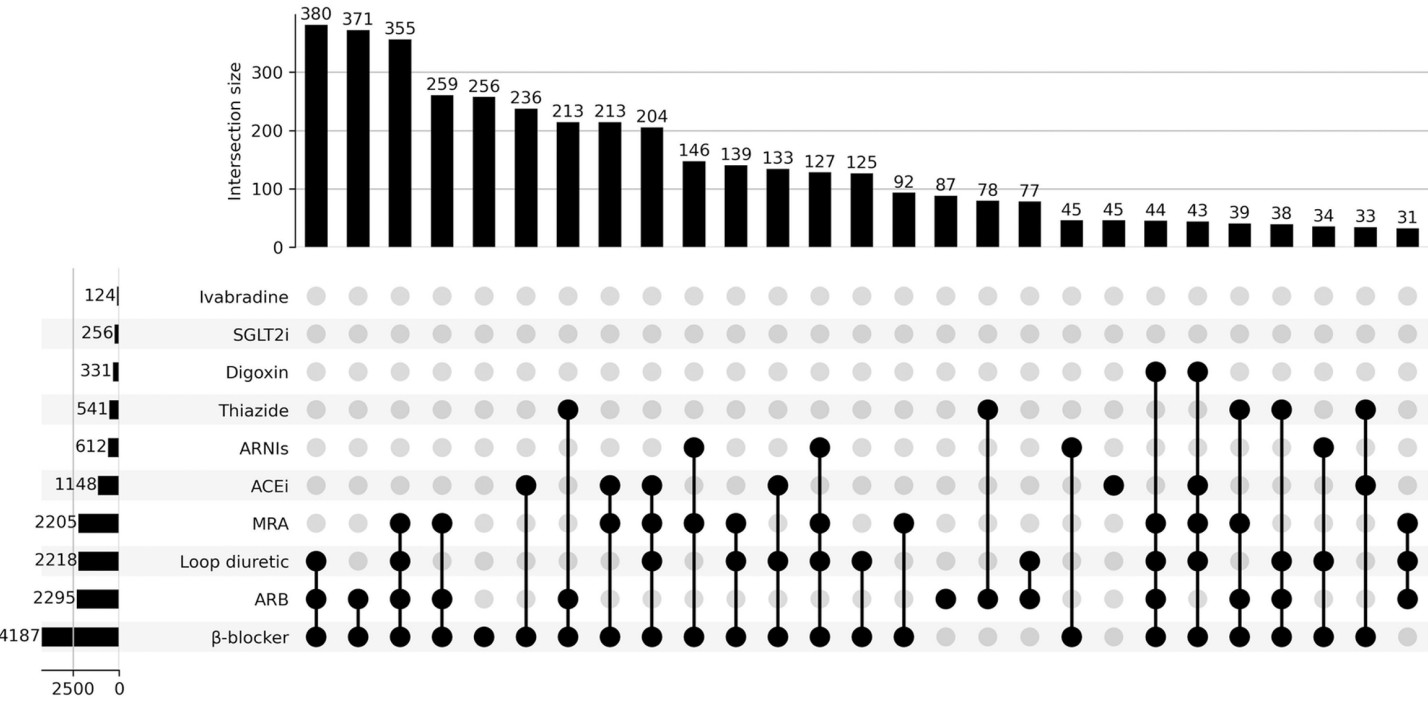

**Fig 1. Main drug therapies used to manage heart failure in the study population (including those combinations or drugs with at least 30 observations).**

with heart failure, creating the need to timely identify and improve the prevention of complications and adverse outcomes [22,23].

When comparing the results with the AMERICANS ASS Registry study, which included 1,000 patients, mainly from Colombia (19.2%), the patients were of similar age and predominantly male. However, they had a lower frequency of hypertension and coronary artery disease, but similar rates of AF and diabetes. Interestingly, they found that the most frequent group, with more than half of the patients (59.5%), were those with reduced failure, similar to this study. Regarding the use of medications, 70.7% were using a β-blocker, 56.8% an ARB, and 30.7% an SGLT2i; however, they were all grouped together without separating by ejection fraction classification. And interestingly, 21.7% received optimal therapy with a β-blocker, ARB, SGLT2i, and an ACEI, ARB, or ARNI [24].

In the management of patients with HFpEF, or HFmrEF some points of interest were identified, such as the use of ARNIs in up to 3.6% and 12.1 respectively of cases and a clear recommendation for this therapy by clinical practice guidelines for patients with HFrEF due to its impact on morbidity and associated mortality [25,26]. The evidence is not conclusive in those whose LVEF is above 40%, but the costs of therapy, as well as the risk of adverse reactions, especially severe hypotension, and their low impact on mortality may make them a therapy that is not so necessary [27,28]. Similarly, in the group of patients with HFpEF, 27.4% used an MRA, which did not have an impact on mortality either but has been associated with endocrine-type adverse reactions and hyperkalemia, although MRAs may reduce the risk of HF hospitalization [29]. Additionally, 4.3% of the patients with HFpEF and with HFmrEF were treated with an SGLT2i during the observation period, which, at the time, did not have any evidence or recommendation for its use; however, in studies published in 2021 and 2022, empagliflozin and dapagliflozin, respectively, showed a significant benefit in the primary composite outcome of hospitalization for heart failure or cardiovascular death, and the results were mainly explained by a

**Table 3. Main concomitant medications received by a group of 4742 patients diagnosed with HF. from Colombia.**

| Comedications | Total (n = 4742) | | Preserved LVEF[a] (HFpEF) (n = 1244) | | Mildly reduced LVEF (HFmrEF) (n = 506) | | Reduced LVEF (HFrEF) (n = 1533) | | No data LVEF (n = 1459) | |
|---|---|---|---|---|---|---|---|---|---|---|
| | n | % | n | % | n | % | n | % | n | % |
| Statins | 3398 | 71.7 | 907 | 72.9 | 392 | 77.5 | 1151 | 75.1 | 948 | 65.0 |
| Aspirin | 2455 | 51.8 | 658 | 52.9 | 267 | 52.8 | 822 | 53.6 | 708 | 48.5 |
| Proton pump inhibitor | 1454 | 30.7 | 363 | 29.2 | 143 | 28.3 | 516 | 33.7 | 432 | 29.6 |
| Anticoagulants[b] | 1223 | 25.8 | 308 | 24.8 | 144 | 28.5 | 437 | 28.5 | 334 | 22.9 |
| Calcium channel blockers (vasoselective) | 1031 | 21.7 | 375 | 30.1 | 100 | 19.8 | 181 | 11.8 | 375 | 25.7 |
| Levothyroxine | 997 | 21.0 | 278 | 22.3 | 111 | 21.9 | 333 | 21.7 | 275 | 18.8 |
| Acetaminophen | 971 | 20.5 | 274 | 22.0 | 104 | 20.6 | 286 | 18.7 | 307 | 21.0 |
| Metformin | 908 | 19.1 | 222 | 17.8 | 97 | 19.2 | 316 | 20.6 | 273 | 18.7 |
| Basal insulin | 532 | 11.2 | 116 | 9.3 | 43 | 8.5 | 196 | 12.8 | 177 | 12.1 |
| Other antiplatelet agents[c] | 451 | 9.5 | 81 | 6.5 | 67 | 13.2 | 176 | 11.5 | 127 | 8.7 |
| Selective Serotonin Reuptake Inhibitors | 344 | 7.3 | 108 | 8.7 | 33 | 6.5 | 97 | 6.3 | 106 | 7.3 |
| Partial agonist opioids | 309 | 6.5 | 94 | 7.6 | 37 | 7.3 | 81 | 5.3 | 97 | 6.6 |
| DPP-4i[d] | 303 | 6.4 | 70 | 5.6 | 26 | 5.1 | 115 | 7.5 | 92 | 6.3 |
| Fast acting insulin | 287 | 6.1 | 64 | 5.1 | 23 | 4.5 | 101 | 6.6 | 99 | 6.8 |
| Anti-arrhythmics | 265 | 5.6 | 50 | 4.0 | 24 | 4.7 | 129 | 8.4 | 62 | 4.2 |
| Aluminium hydroxide | 243 | 5.1 | 75 | 6.0 | 19 | 3.8 | 75 | 4.9 | 74 | 5.1 |
| Ranitidine | 234 | 4.9 | 55 | 4.4 | 26 | 5.1 | 88 | 5.7 | 65 | 4.5 |
| Alpha blockers | 218 | 4.6 | 61 | 4.9 | 26 | 5.1 | 60 | 3.9 | 71 | 4.9 |
| 1st-generation antihistamines | 150 | 3.2 | 47 | 3.8 | 12 | 2.4 | 41 | 2.7 | 50 | 3.4 |
| Clonidine | 126 | 2.7 | 44 | 3.5 | 13 | 2.6 | 26 | 1.7 | 43 | 2.9 |
| Antiepileptics | 124 | 2.6 | 34 | 2.7 | 17 | 3.4 | 39 | 2.5 | 34 | 2.3 |
| Tricyclic Antidepressants | 122 | 2.6 | 26 | 2.1 | 12 | 2.4 | 46 | 3.0 | 38 | 2.6 |
| Total agonist opioids | 101 | 2.1 | 28 | 2.3 | 13 | 2.6 | 29 | 1.9 | 31 | 2.1 |
| Fibrates | 99 | 2.1 | 25 | 2.0 | 16 | 3.2 | 26 | 1.7 | 32 | 2.2 |
| GLP-1 analogs[e] | 80 | 1.7 | 18 | 1.4 | 7 | 1.4 | 27 | 1.8 | 28 | 1.9 |
| Ezetimibe | 79 | 1.7 | 18 | 1.4 | 14 | 2.8 | 34 | 2.2 | 13 | 0.9 |
| Glucocorticoids | 78 | 1.6 | 18 | 1.4 | 12 | 2.4 | 24 | 1.6 | 24 | 1.6 |
| Benzodiazepines | 76 | 1.6 | 20 | 1.6 | 9 | 1.8 | 23 | 1.5 | 24 | 1.6 |
| Sucralfate | 63 | 1.3 | 11 | 0.9 | 10 | 2.0 | 21 | 1.4 | 21 | 1.4 |
| Non-steroidal anti-inflammatory drugs | 63 | 1.3 | 12 | 1.0 | 5 | 1.0 | 22 | 1.4 | 24 | 1.6 |
| Atypical antipsychotic | 60 | 1.3 | 13 | 1.0 | 7 | 1.4 | 18 | 1.2 | 22 | 1.5 |
| Hyoscine butylbromide | 50 | 1.1 | 10 | 0.8 | 4 | 0.8 | 21 | 1.4 | 15 | 1.0 |
| Calcium channel blockers (cardiodepressants) | 45 | 0.9 | 20 | 1.6 | 3 | 0.6 | 5 | 0.3 | 17 | 1.2 |
| Typical antipsychotic | 38 | 0.8 | 12 | 1.0 | 3 | 0.6 | 12 | 0.8 | 11 | 0.8 |
| 2nd generation antihistamine | 30 | 0.6 | 8 | 0.6 | 2 | 0.4 | 10 | 0.7 | 10 | 0.7 |
| Antiparkinsonians | 26 | 0.5 | 7 | 0.6 | 2 | 0.4 | 5 | 0.3 | 12 | 0.8 |
| Memantine | 18 | 0.4 | 6 | 0.5 | 2 | 0.4 | 3 | 0.2 | 7 | 0.5 |
| Sulfonylureas | 15 | 0.3 | 4 | 0.3 | 1 | 0.2 | 4 | 0.3 | 6 | 0.4 |
| Serotonin and norepinephrine reuptake inhibitors | 14 | 0.3 | 4 | 0.3 | 2 | 0.4 | 6 | 0.4 | 2 | 0.1 |
| Z Drugs | 7 | 0.1 | 2 | 0.2 | 1 | 0.2 | 1 | 0.1 | 3 | 0.2 |
| Antidementia | 7 | 0.1 | 1 | 0.1 | 2 | 0.4 | 1 | 0.1 | 3 | 0.2 |
| Omega-3 acids | 2 | 0.0 | 1 | 0.1 | 0 | 0.0 | 0 | 0.0 | 1 | 0.1 |

*(Continued)*

**Table 3.** (Continued)

| Comedications | Total (n = 4742) | | Preserved LVEF[a] (HFpEF) (n = 1244) | | Mildly reduced LVEF (HFmrEF) (n = 506) | | Reduced LVEF (HFrEF) (n = 1533) | | No data LVEF (n = 1459) | |
|---|---|---|---|---|---|---|---|---|---|---|
| | n | % | n | % | n | % | n | % | n | % |
| Dipyrone | 1 | 0.0 | 1 | 0.1 | 0 | 0.0 | 0 | 0.0 | 0 | 0.0 |

[a]LVEF: left ventricular ejection fraction.

[b]Includes orals and heparins.

[c]Clopidogrel. prasugrel. and ticagrelor.

[d]DPP-4: dipeptidyl peptidase 4 inhibitors.

[e]GLP-1: glucagon-like peptide 1.

significant reduction in hospitalizations but without a clear benefit in mortality [30,31]. Additionally, as found in this analysis by studying not only the use of medications but also the reasons for hospitalization in depth, hospitalization was not only due to HF, but a significant proportion (approximately 40%) of individuals were hospitalized for any other cause, indicating the importance of providing multidisciplinary management and a comprehensive approach, including an optimal approach to secondary cardiovascular prevention, arrhythmias, diabetes mellitus, or chronic obstructive pulmonary disease, which could be a therapeutic strategy better than the use of therapies with a benefit/risk balance that is not currently clear in HFpEF [32,33]. It is important to identify new therapeutic options with an impact on morbidity and with a good safety profile and that may even have some impact on cardiovascular mortality [34,35].

The assessment of the treatment prescribed to patients with HFrEF identified that 85% received RAASis, and only 26.4% received ARNIs, which are currently the recommended first-line therapy due to their impact on survival, hospitalization, and quality of life compared to other available treatments [7,13,25,36]. During the observation period of this study, ARNIs were an alternative to ACEIs, although they are already used in similar proportions as the other two therapies, which translates into a progressive adoption of the recommendation based on published evidence at the time. A total of 7.0% of patients with reduced LVEF used SGLT2is, despite the recommendations made in the clinical practice guidelines from studies with empagliflozin and dapagliflozin in the management of HF starting in 2021 [14,15], optimizing the pharmacological treatment of these patients.

The study by Störk et al. found that patients with better functional classes (I and II) according to the NYHA had 93% and 55% compliance, respectively, with the optimal therapy according to the recommendations of the clinical practice guidelines, while this was reduced to 21% when they had functional classes III and IV, although the patients were not classified by LVEF [16]. A study conducted by Rastogi et al. in France between 2015 and 2019 found suboptimal management in approximately 70% of patients [37], especially a lack of prescriptions of MRAs and ARNIs for patients with HFrEF, similar to what was found in Colombia, where among patients with reduced LVEF, only 58.1% used the recommended triple therapy (RAASis + β-blockers+MRAs) and only 4.9% also used an SGLT2i, showing a potential possibility of optimizing drug treatment according to current recommendations [11]. According to the meta-analysis by Vaduganathan et al. in 2020, the combination of an ARNI, an SGLT2i, an MRA and a β-blocker can achieve up to 8.3 years of cardiovascular survival time and up to 6.3 years of additional survival time compared to conventional ACEI treatment plus a β-blocker [38], constituting a therapy based on pathophysiology and the best evidence with a significant impact on survival, hospitalization rates, symptoms and quality of life. In addition, different authors have recently identified that the unique clinical profiles of patients with heart failure may partly explain their failure to comply with the recommendations of the clinical practice guidelines, mainly explained by low drug tolerability caused by severe hypotension, bradycardia, altered renal function or hyperkalemia [39–42]. According to these situations, in 2021, the European Society of Cardiology defined,

through expert consensus, nine patient profiles that may be relevant for the therapeutic management of HFrEF, including heart rate ranges, presence of atrial fibrillation, symptomatic hypotension, impaired renal function, and presence of hyperkalemia in the differentiating profile, thus showing the importance of offering personalized management of heart failure and adjusting the optimal therapy recommended according to clinical practice guidelines to the profile of the patient to achieve a more comprehensive management of the disease [43].

Different studies carried out among patients with heart failure, such as EMPEROR that evaluated empagliflozin in the management of subjects with preserved LVEF [35] or the report by Rastogi et al. [37], showed that patients received β-blockers in cases of reduced LVEF in proportions greater than 80% and 90%, respectively, which is comparable with the findings of this analysis. However, the usefulness of this therapy is currently being discussed in cases with preserved LVEF due to insufficient evidence to support its impact on clinical outcomes, added to the possibility of altering central arterial pressure, increasing wall stress, and exacerbating chronotropic incompetence; therefore, symptoms may worsen without the benefit shown in patients with reduced LVEF [44]. Enriching the discussion regarding the responsible use of β-blockers for the appropriate indications can enhance rationality in the prescription and optimization of resources.

This study has some limitations that must be taken into account. First, due to its observational nature and the secondary source of information from electronic clinical records, including the lack of data such as LVEF or GFR in a portion of patients, it was difficult to perform certain evaluations in this study, but this study did show the reality of medical care in a real-life context and the follow-up that is provided to patients with heart failure in Colombia. There have been no reports of other paraclinical tests of interest, and the information on hospital care was insufficient. It can be assumed that some patients may be misclassified and a significant proportion were low risk, being classified as stage B or stage C functional class I, still showing the reality of patients under treatment. The strengths include the large sample size for an observational study with a systematic review of medical records; this was one of the largest studies conducted in Colombia with coverage of the main regions of the country and the complete identification of treatment patterns according to the clinical categories of heart failure.

## Conclusions

In this study, the cohort of patients with heart failure in Colombia mainly comprised men, had a mean age of 68 years, mostly had preserved LVEF and stage C, had HF caused by ischemic heart disease or arterial hypertension and were also affected by obesity, diabetes mellitus, or atrial fibrillation; many participants experienced dyspnea on exertion, anginal pain, and lower limb edema, leading tso hospitalization in the last year, and were primarily treated with β-blockers, RAASis, especially ARBs, and a low proportion of ARNIs. However, the treatment that patients with HFpEF and HFmrEF receive is simpler, as fewer medications are used and the control of comorbidities is prioritized. Thus, it was found that the current therapy for these groups in this study more similar to the recommendations made by clinical practice guidelines and current evidence, while for those with HFrEF, the proportion of therapies according to these recommendations is lower, which provides an important opportunity to optimize and personalize management and therefore surely improve clinical outcomes.

## Acknowledgments

**We thank Ana Maria Baena, Vanessa Parra, Julio Cesar Arce and Harrison Ospina for their work in obtaining the database. Marcela Rivera for your support in discussion or results.**

**Consent to participate**. No applicable, is a retrospective observational study

**Consent to publish**: all authors consent to participate

## Author contributions

**Conceptualization:** Juan Sebastián Franco, María de Rosario Forero, David Vizcaya, Jorge E. Machado-Alba.

**Data curation:** Manuel E Machado-Duque, Andrés Gaviria-Mendoza.

**Formal analysis:** Manuel E Machado-Duque, Andrés Gaviria-Mendoza, Luis F Valladales-Restrepo, Jorge E. Machado-Alba.

**Funding acquisition:** Juan Sebastián Franco, María de Rosario Forero, David Vizcaya, Jorge E. Machado-Alba.

**Investigation:** Manuel E Machado-Duque, Andrés Gaviria-Mendoza, Jorge E. Machado-Alba.

**Methodology:** Manuel E Machado-Duque, Andrés Gaviria-Mendoza, Luis F Valladales-Restrepo, Jorge E. Machado-Alba.

**Project administration:** Jorge E. Machado-Alba.

**Supervision:** Jorge E. Machado-Alba.

**Validation:** Luis F Valladales-Restrepo.

**Visualization:** Juan Sebastián Franco, María de Rosario Forero, David Vizcaya.

**Writing – original draft:** Manuel E Machado-Duque.

**Writing – review & editing:** Jorge E. Machado-Alba.

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
