## [Decision Letter · Decision Letter 0]

PONE-D-25-02108HEArt failure Treatment patterns: A pharmacoepidemiological descriptive study in COlombia (the HEATCO study)PLOS ONE

Dear Dr. Machado Alba,

Thank you for submitting your manuscript to PLOS ONE. After careful consideration, we feel that it has merit but does not fully meet PLOS ONE’s publication criteria as it currently stands. Therefore, we invite you to submit a revised version of the manuscript that addresses the points raised during the review process.

The study's strengths include its large sample size and detailed design, but concerns about population representation and LVEF classification suggest a need for reevaluation of conclusions and inclusion of broader Latin American data comparisons.

We look forward to receiving your revised manuscript.

Kind regards,

Francesco Curcio, M.D., Ph.D.

Academic Editor

PLOS ONE

Journal Requirements:

2. Thank you for stating the following financial disclosure: [This work was supported by Bayer AG, (Bogotá, Colombia)]. 

Please include this amended Role of Funder statement in your cover letter; we will change the online submission form on your behalf."

3.  Thank you for stating the following in the Competing Interests section: [Manuel Machado-Duque have a contractual relationship with Audifarma SA and Institución Universitaria Visión de las Américas.  Andres Gaviria-Mendoza have a contractual relationship with Audifarma SA and Institución Universitaria Visión de las Américas. Luis Valladales-Restrepo have a contractual relationship with Audifarma SA and Institución Universitaria Visión de las Américas. Juan-Sebastian Franco are full-time employee of Bayer Colombia. Maria del Rosario Forero are full-time employee of Bayer Colombia. David Vizcaya are full-time employee of Bayer Hispania (Spain). Marcela Rivera are full-time employee of Bayer Hispania (Spain). Jorge Machado-Alba have a contractual relationship with Universidad Tecnológica de Pereira and Audifarma SA.].

Reviewers' comments:

Reviewer's Responses to Questions

**Comments to the Author**

1. Is the manuscript technically sound, and do the data support the conclusions?

Reviewer #1: Partly

Reviewer #2: Yes

2. Has the statistical analysis been performed appropriately and rigorously? 

Reviewer #1: No

Reviewer #2: Yes

3. Have the authors made all data underlying the findings in their manuscript fully available?

Reviewer #1: Yes

Reviewer #2: Yes

4. Is the manuscript presented in an intelligible fashion and written in standard English?

Reviewer #1: Yes

Reviewer #2: Yes

5. Review Comments to the Author

Reviewer #1: Dr Machado Alba et al present a interesting manuscript that evaluated the prescription patterns of medications for heart failure in a cohort of 4742 patients from Colombia, in a a retrospective

study based on the clinical records of patients. The main findings were: high use of betablockers, RAAS inhibitors and MRAs, intermediate prescription of ARNI y low use of SGLT2 inhibitors, triple therapy with RAASis+β-blockers+MRAs in 58.1% of patients with reduced LVEF and 25.2% with preserved LVEF, while quadruple therapy was used in less than 5%. Authors concluded that subjects with HF and preserved LVEF were treated closer to the recommendations of clinical practice guidelines, while the proportion of indicated therapies according to guidelines recommendations is lower among those with reduced LVEF.

Among the strengths of the study are the fact of presenting real-world information about drugs prescription, the huge sample size, and a very comprehensive design of the protocol. However, there are some comments to evaluate before to being accepted.

Abstract:

1. Authors should provide in this section data about classification of HF according with LVEF, because the definition used in the study (reduced or preserved) is different from the currently accepted (see comment below). In addition, the proportion of patients without LVEF available must be described here.

2. The conclusion “The treatment that patients with heart failure with preserved LVEF receive is closer to the recommendations of clinical practice guidelines” is confused and probably misinterpreted. First, the recommendations of treatment for HFpEF in 2020 were only diuretics and treating comorbidities. Second, current recommendation are SGLT2i and diuretics, with lower level of evidence MRA, ARB and ARNI. Moreover, in this study the proportion of combination therapy was higher in HFrEF. So, this conclusion should be reconsidered

Material and methods

3. One concern is the population included in this study. Age of 68.2 years, no data about LVEF in 30.7% (n=1459), 6,7% in stage B, 54.8% en FC I, only 16% of cardiomyopathies (19% in LVEF below 40), 58.7% of those with LVEF measurements with values above 40%, 5.8% of hospitalizations, and only 6.4% treated by cardiologist suggest that it was at least low risk population, and some patients could be misclassified as HF. This patient profile might be associated with the treatment, since the evidence in patients in stage B or in FC I is different from those in Stage C and FC II and higher. These limitations must be discussed in limitations sections.

4. A major concern is the classification by LVEF. According with Universal definition of HF and all major guidelines, preserved HF is defined by an value ≥50%, and values 41-49 is considered HF with mildly reduced EF (HFmrEF). Treatment recommendation vary between these 2 groups. I suggest reclassified by EF in HFrEF, HFmrEF and HFpEF, presenting in the table the statistical comparison among 3 groups in drug prescription.

Discussion

5. It is important compare the study results not only with European or NorthAmerican cohorts, but with LatinAmerican data, from different countries as well as with AMERICCAASS Registry (Clin Cardiol. 2024 Feb;47(2):e24182. doi: 10.1002/clc.24182. Epub 2023 Nov 30) and global registries that include data from LA (REPORT-HF and INTER-CHF)

6. Again, the conclusion “The treatment that patients with heart failure with preserved LVEF receive is closer to the recommendations of clinical practice guidelines” is confused and probably misinterpreted. First, the recommendations of treatment for HFpEF in 2020 were only diuretics and treating comorbidities. Second, current recommendation are SGLT2i and diuretics, with lower level of evidence MRA, ARB and ARNI. Moreover, in this study the proportion of combination therapy was higher in HFrEF. So, this conclusion should be reconsidered

Reviewer #2: The authors retrospectively examine the prescription patterns of medications for the treatment of heart failure in a cohort of patients from Colombia between 2019 and 2020. Patients were classified according to functional class, stage, and left ventricular ejection fraction (LVEF). A total of 4742 patients were evaluated, with a mean age of 68.2±13.8 years and a male predominance (61.3%). A total of 92.0% were classified as stage C and 54.8% as functional class I, the mean LVEF was 42.9±14.8%, and 28.5% had reduced LVEF. The most common causes were ischemic heart disease (44.0%) and arterial hypertension (29.7%). A total of 5.2% had hospitalizations for heart failure in the last year, and 75.6% were attended by a general practitioner. These patients were treated with β-blockers (88.3%), renin-angiotensin-aldosterone system inhibitors (RAASis) (83.1%), loop diuretics (46.8%), and mineralocorticoid receptor antagonists (MRAs) (46.5%). Triple therapy with RAASis+β-blockers+MRAs was received by 58.1% of patients with reduced LVEF and 25.2% with preserved LVEF, while quadruple therapy adding a sodium-glucose cotransporter-2 inhibitor (SGLT2i) was given to 4.9% and 1.4%, respectively.

The review is interesting, but the mean problem is the lack of consideration of the condition of frailty which represents one of the most frequent conditions affecting patients with heart failure.

Please consider and discuss:

Testa G, Curcio F, Liguori I, Basile C, Papillo M, Tocchetti CG, Galizia G, Della-Morte D, Gargiulo G, Cacciatore F, Bonaduce D, Abete P. Physical vs. multidimensional frailty in older adults with and without heart failure. ESC Heart Fail. 2020 Jun;7(3):1371-1380.

Cacciatore F, Amarelli C, Maiello C, Pratillo M, Tosini P, Mattucci I, Salerno G, Curcio F, Elia F, Mercurio V, Golino P, Bonaduce D, Abete P. Effect of Sacubitril-Valsartan in reducing depression in patients with advanced heart failure. J Affect Disord. 2020 Jul 1;272:132-137.

6. PLOS authors have the option to publish the peer review history of their article (what does this mean? ). If published, this will include your full peer review and any attached files.

**Do you want your identity to be public for this peer review?** For information about this choice, including consent withdrawal, please see our Privacy Policy .

Reviewer #1: **Yes: ** Eduardo R Perna, MD

Reviewer #2: No

---

## [Author Response · Author response to Decision Letter 1]

21 Apr 2025

PONE-D-25-02108

HEArt failure Treatment patterns: A pharmacoepidemiological descriptive study in COlombia (the HEATCO study)

Dear Francesco Curcio, M.D., Ph.D.

Academic Editor

PLOS ONE

We have addressed each of the points highlighted by the reviewers and are confident that the changes will improve the quality of the work.

We include the following items:

Journal Requirements:

2. Thank you for stating the following financial disclosure: [This work was supported by Bayer AG, (Bogotá, Colombia)].

Please state what role the funders took in the study. If the funders had no role, please state: "The funders had no role in study design, data collection and analysis, decision to publish, or preparation of the manuscript.""

R/ we adjust.

3. Thank you for stating the following in the Competing Interests section: [Manuel Machado-Duque have a contractual relationship with Audifarma SA and Institución Universitaria Visión de las Américas. Andres Gaviria-Mendoza have a contractual relationship with Audifarma SA and Institución Universitaria Visión de las Américas. Luis Valladales-Restrepo have a contractual relationship with Audifarma SA and Institución Universitaria Visión de las Américas. Juan-Sebastian Franco are full-time employee of Bayer Colombia. Maria del Rosario Forero are full-time employee of Bayer Colombia. David Vizcaya are full-time employee of Bayer Hispania (Spain). Marcela Rivera are full-time employee of Bayer Hispania (Spain). Jorge Machado-Alba have a contractual relationship with Universidad Tecnológica de Pereira and Audifarma SA.].

R/ we include “This does not alter our adherence to PLOS ONE policies on sharing data and materials”.

R/ Competing Interests statement is included in cover letter.

Reviewers' comments:

Reviewer's Responses to Questions

Comments to the Author

1. Is the manuscript technically sound, and do the data support the conclusions?

Reviewer #1: Partly

Reviewer #2: Yes

2. Has the statistical analysis been performed appropriately and rigorously?

Reviewer #1: No

Reviewer #2: Yes

3. Have the authors made all data underlying the findings in their manuscript fully available?

Reviewer #1: Yes

Reviewer #2: Yes

4. Is the manuscript presented in an intelligible fashion and written in standard English?

Reviewer #1: Yes

Reviewer #2: Yes

5. Review Comments to the Author

Reviewer #1: Dr Machado-Alba et al present a interesting manuscript that evaluated the prescription patterns of medications for heart failure in a cohort of 4742 patients from Colombia, in a retrospective study based on the clinical records of patients. The main findings were: high use of betablockers, RAAS inhibitors and MRAs, intermediate prescription of ARNI y low use of SGLT2 inhibitors, triple therapy with RAASis+β-blockers+MRAs in 58.1% of patients with reduced LVEF and 25.2% with preserved LVEF, while quadruple therapy was used in less than 5%. Authors concluded that subjects with HF and preserved LVEF were treated closer to the recommendations of clinical practice guidelines, while the proportion of indicated therapies according to guidelines recommendations is lower among those with reduced LVEF.

Among the strengths of the study are the fact of presenting real-world information about drugs prescription, the huge sample size, and a very comprehensive design of the protocol. However, there are some comments to evaluate before to being accepted.

Abstract:

1. Authors should provide in this section data about classification of HF according with LVEF, because the definition used in the study (reduced or preserved) is different from the currently accepted (see comment below). In addition, the proportion of patients without LVEF available must be described here.

R/ The % of patients without data is included

2. The conclusion “The treatment that patients with heart failure with preserved LVEF receive is closer to the recommendations of clinical practice guidelines” is confused and probably misinterpreted. First, the recommendations of treatment for HFpEF in 2020 were only diuretics and treating comorbidities. Second, current recommendation are SGLT2i and diuretics, with lower level of evidence MRA, ARB and ARNI. Moreover, in this study the proportion of combination therapy was higher in HFrEF. So, this conclusion should be reconsidered

R/ is adjusted to improve interpretation

Material and methods

3. One concern is the population included in this study. Age of 68.2 years, no data about LVEF in 30.7% (n=1459), 6,7% in stage B, 54.8% en FC I, only 16% of cardiomyopathies (19% in LVEF below 40), 58.7% of those with LVEF measurements with values above 40%, 5.8% of hospitalizations, and only 6.4% treated by cardiologist suggest that it was at least low risk population, and some patients could be misclassified as HF. This patient profile might be associated with the treatment, since the evidence in patients in stage B or in FC I is different from those in Stage C and FC II and higher. These limitations must be discussed in limitations sections.

R/ is a real-life study of patients diagnosed and treated for heart failure. It shows, among many things, that some patients do not see specialists or have echocardiogram reports. It expands the discussion of limitations.

4. A major concern is the classification by LVEF. According with Universal definition of HF and all major guidelines, preserved HF is defined by an value ≥50%, and values 41-49 is considered HF with mildly reduced EF (HFmrEF). Treatment recommendation vary between these 2 groups. I suggest reclassified by EF in HFrEF, HFmrEF and HFpEF, presenting in the table the statistical comparison among 3 groups in drug prescription.

R/ is reclassified to be in line with the current classification. Between reduced, mildly reduced, or preserved.

Discussion

5. It is important compare the study results not only with European or NorthAmerican cohorts, but with LatinAmerican data, from different countries as well as with AMERICCAASS Registry (Clin Cardiol. 2024 Feb;47(2):e24182. doi: 10.1002/clc.24182. Epub 2023 Nov 30) and global registries that include data from LA (REPORT-HF and INTER-CHF)

R/ When the manuscript was initially written, the results of these were not available, nor were they published. They are being reviewed and the discussion is being expanded with these (reference 24).

6. Again, the conclusion “The treatment that patients with heart failure with preserved LVEF receive is closer to the recommendations of clinical practice guidelines” is confused and probably misinterpreted. First, the recommendations of treatment for HFpEF in 2020 were only diuretics and treating comorbidities. Second, current recommendation are SGLT2i and diuretics, with lower level of evidence MRA, ARB and ARNI. Moreover, in this study the proportion of combination therapy was higher in HFrEF. So, this conclusion should be reconsidered

R/ The wording of the conclusion is adjusted

Reviewer #2: The authors retrospectively examine the prescription patterns of medications for the treatment of heart failure in a cohort of patients from Colombia between 2019 and 2020. Patients were classified according to functional class, stage, and left ventricular ejection fraction (LVEF). A total of 4742 patients were evaluated, with a mean age of 68.2±13.8 years and a male predominance (61.3%). A total of 92.0% were classified as stage C and 54.8% as functional class I, the mean LVEF was 42.9±14.8%, and 28.5% had reduced LVEF. The most common causes were ischemic heart disease (44.0%) and arterial hypertension (29.7%). A total of 5.2% had hospitalizations for heart failure in the last year, and 75.6% were attended by a general practitioner. These patients were treated with β-blockers (88.3%), renin-angiotensin-aldosterone system inhibitors (RAASis) (83.1%), loop diuretics (46.8%), and mineralocorticoid receptor antagonists (MRAs) (46.5%). Triple therapy with RAASis+β-blockers+MRAs was received by 58.1% of patients with reduced LVEF and 25.2% with preserved LVEF, while quadruple therapy adding a sodium-glucose cotransporter-2 inhibitor (SGLT2i) was given to 4.9% and 1.4%, respectively.

The review is interesting, but the mean problem is the lack of consideration of the condition of frailty which represents one of the most frequent conditions affecting patients with heart failure.

Please consider and discuss:

Testa G, Curcio F, Liguori I, Basile C, Papillo M, Tocchetti CG, Galizia G, Della-Morte D, Gargiulo G, Cacciatore F, Bonaduce D, Abete P. Physical vs. multidimensional frailty in older adults with and without heart failure. ESC Heart Fail. 2020 Jun;7(3):1371-1380.

Cacciatore F, Amarelli C, Maiello C, Pratillo M, Tosini P, Mattucci I, Salerno G, Curcio F, Elia F, Mercurio V, Golino P, Bonaduce D, Abete P. Effect of Sacubitril-Valsartan in reducing depression in patients with advanced heart failure. J Affect Disord. 2020 Jul 1;272:132-137.

R/ discussion on fragility is included (reference 22)________________________________________

6. PLOS authors have the option to publish the peer review history of their article (what does this mean?). If published, this will include your full peer review and any attached files.

R/ No

1. Please upload a Response to Reviewers letter which should include a point by point

response to each of the points made by the Editor and / or Reviewers. (This should be

uploaded as a 'Response to Reviewers' file type.) Please follow this link for more

information: http://blogs.PLOS.org/everyone/2011/05/10/how-to-submit-your-revisedmanuscript/

R/ We include a response to reviewers letter

2. Can you please upload an additional copy of your revised manuscript that does not

contain any tracked changes or highlighting as your main article file. This will be used in

the production process if your manuscript is accepted. Please amend the file type for the

file showing your changes to Revised Manuscript w/tracked changes. Please follow this

link for more information: http://blogs.PLOS.org/everyone/2011/05/10/how-to-submit-yourrevised-

manuscript/

R/ We include a clean manuscript file

3. Thank you for uploading your study's underlying data set. Unfortunately, the repository

you have noted in your Data Availability statement does not qualify as an acceptable data

repository according to PLOS ONE's standards.

R/ The protocolos.io repository is public and open access and is now available at:

https://www.protocols.io/private/0031A1C4379F11EE93150A58A9FEAC02

Although it appears as private to reviewers and is currently available, once accepted for publication it will be freely available to any researcher or individual who wishes to access it.

The authors

---

## [Decision Letter · Decision Letter 1]

HEArt failure Treatment patterns: A pharmacoepidemiological descriptive study in COlombia (the HEATCO study)

PONE-D-25-02108R1

Dear Dr. Jorge Enrique Machado Alba ,

We’re pleased to inform you that your manuscript has been judged scientifically suitable for publication and will be formally accepted for publication once it meets all outstanding technical requirements.

Kind regards,

Francesco Curcio, M.D., Ph.D.

Academic Editor

PLOS ONE

Additional Editor Comments (optional):

In accordance with the reviewers’ recommendations, we are pleased to accept the manuscript.

Reviewers' comments:

Reviewer's Responses to Questions

**Comments to the Author**

1. If the authors have adequately addressed your comments raised in a previous round of review and you feel that this manuscript is now acceptable for publication, you may indicate that here to bypass the “Comments to the Author” section, enter your conflict of interest statement in the “Confidential to Editor” section, and submit your "Accept" recommendation.

Reviewer #1: All comments have been addressed

Reviewer #2: (No Response)

2. Is the manuscript technically sound, and do the data support the conclusions?

Reviewer #1: (No Response)

Reviewer #2: Yes

3. Has the statistical analysis been performed appropriately and rigorously? 

Reviewer #1: (No Response)

Reviewer #2: Yes

4. Have the authors made all data underlying the findings in their manuscript fully available?

Reviewer #1: (No Response)

Reviewer #2: Yes

5. Is the manuscript presented in an intelligible fashion and written in standard English?

Reviewer #1: (No Response)

Reviewer #2: (No Response)

6. Review Comments to the Author

Reviewer #1: This manuscript was previously reviewed and has been revised. The revised paper and the response of the Authors to the prior reviews were examined. The paper was summarized in my earlier review. The comments offered in my initial review have been addressed in a satisfactory manner by the authors. There was a useful modification of the manuscript and I recommend accepting it

Reviewer #2: No further comments. The manuscript is really improved, all questions are solved and the manuscript merits to be published.

7. PLOS authors have the option to publish the peer review history of their article (what does this mean? ). If published, this will include your full peer review and any attached files.

**Do you want your identity to be public for this peer review?** For information about this choice, including consent withdrawal, please see our Privacy Policy .

Reviewer #1: No

Reviewer #2: No

---

## [Editor Report · Acceptance letter]

PONE-D-25-02108R1

PLOS ONE

Dear Dr. Machado-Alba,

I'm pleased to inform you that your manuscript has been deemed suitable for publication in PLOS ONE. Congratulations! Your manuscript is now being handed over to our production team.

Kind regards,

on behalf of

Dr. Francesco Curcio

Academic Editor

PLOS ONE